# Long-Term Conditions Questionnaire (LTCQ): initial validation survey among primary care patients and social care recipients in England

Caroline M Potter,[1,2] Laurie Batchelder,[3,4] Christine A'Court,[5] Louise Geneen,[1,4] Laura Kelly,[1,2] Diane Fox,[3,4] Matthew Baker,[4] Jennifer Bostock,[4] Angela Coulter,[1,4] Ray Fitzpatrick,[1,4,2] Julien E Forder,[3,4] Elizabeth Gibbons,[1,2] Crispin Jenkinson,[1,4] Karen Jones,[3,4] Michele Peters[1,4]

[1]Health Services Research Unit, Nuffield Department of Population Health, University of Oxford, Oxford, UK
[2]NIHR Collaboration for Leadership in Applied Health Research and Care (CLAHRC) Oxford, Oxford, UK
[3]Personal Social Services Research Unit, School of Social Policy Sociology and Social Research, University of Kent, Canterbury, Kent, UK
[4]Department of Health, Quality and Outcomes of Person-centred Care Policy Research Unit, Canterbury, Kent, UK
[5]Nuffield Department of Primary Care Health Sciences, University of Oxford, Oxford, UK

Correspondence to
Dr Caroline M Potter;
caroline.potter@dph.ox.ac.uk

## ABSTRACT

**Objective** The aim of this study was to validate a new generic patient-reported outcome measure, the Long-Term Conditions Questionnaire (LTCQ), among a diverse sample of health and social care users in England.

**Design** Cross-sectional validation survey. Data were collected through postal surveys (February 2016–January 2017). The sample included a healthcare cohort of patients recruited through primary care practices, and a social care cohort recruited through local government bodies that provide social care services.

**Participants** 1211 participants (24% confirmed social care recipients) took part in the study. Healthcare participants were recruited on the basis of having one of 11 specified long-term conditions (LTCs), and social care participants were recruited on the basis of receiving social care support for at least one LTC. The sample exhibited high multimorbidity, with 93% reporting two or more LTCs and 43% reporting a mental health condition.

**Outcome measures** The LTCQ's construct validity was tested with reference to the EQ-5D (5-level version), the Self-Efficacy for Managing Chronic Disease scale, an Activities of Daily Living scale and the Bayliss burden of morbidity scale.

**Results** Low levels of missing data for each item indicate acceptability of the LTCQ across the sample. The LTCQ exhibits high internal consistency (Cronbach's $\alpha$=0.95) across the scale's 20 items and excellent test–retest reliability (intraclass correlation coefficient=0.94, 95% CI 0.93 to 0.95). Associations between the LTCQ and all reference measures were moderate to strong and in the expected directions, indicating convergent construct validity.

**Conclusions** This study provides evidence for the reliability and validity of the LTCQ, which has potential for use in both health and social care settings. The LTCQ could meet a need for holistic outcome measurement that goes beyond symptoms and physical function, complementing existing measures to capture fully what it means to live well with LTCs.

## Strengths and limitations of this study

► This study is the first psychometric evaluation of the Long-Term Conditions Questionnaire, a new generic patient-reported outcome measure.
► The study included a large survey validation sample of 1211 health and social care users in England.
► The sample was highly diverse in terms of illness burden and care needs, with 93% reporting multimorbidity and 43% reporting a mental health condition.
► A limitation was the low response rate achieved among social care users, although this was consistent with previous studies.
► Further validation work among other ethnic groups and in non-English contexts is required since the vast majority of participants were white British.

of health systems worldwide. Considering the high global burden of long-term conditions (LTCs), their management is a top priority in policy terms.[1 2] In England, around one-quarter of the population lives with at least one LTC, accounting for nearly three-quarters of the cost of health and social care services.[3] In particular, the rise of multimorbidity has highlighted the need for integrated services that can address a person's overall health status and enhance their capacity for living well with their conditions.

Patient-reported outcome measures (PROMs) are essential tools for capturing the impact of illness as experienced by those living with LTCs, and the potential of PROMs for facilitating person-centred care has been recognised for more than a decade.[4] However, there is currently no agreed patient-reported measure for evaluating the intended outcome of person-centred care, which could broadly be described as 'living well' within the overall

## INTRODUCTION

Long-term chronic conditions have emerged as a significant challenge to the sustainability

context of one's health condition(s). 'Living well with LTCs' is a complex construct that encompasses both traditional health-related quality of life domains (eg, symptom severity, physical and social functioning, emotional well-being) and more recently recognised domains of importance (eg, treatment burden, empowerment, confidence in self-management).[5 6] A comprehensive measure for assessing the impacts of LTCs should thus move beyond traditional health-status domains of PROMs to capture a more holistic notion of living well.[7]

While a plethora of condition-specific and generic PROMs exist, both types of measure have shortcomings for capturing what it means to live well with LTCs. In focusing on a single disease category, condition-specific PROMs cannot adequately address the phenomenon of multimorbidity, where impacts may be cumulative or interdependent across all conditions.[8] Standardised generic PROMs such as the EQ-5D[9] and SF-36,[10] while valuable for comparative population-level analyses, are unlikely to capture all issues of importance for people living with LTCs.[11 12] Furthermore, these measures may not be appropriate for long-term monitoring of LTCs, where the objectives of health and social care services may be to maintain well-being and/or to avoid deterioration rather than to achieve major health gains.[13] A further complication arises with the distinction between health-related and social care-related measures,[14] as people with complex needs will potentially draw on both types of services. A measure that is appropriate for both contexts could facilitate the development of person-centred care pathways, which are increasingly recognised as preferable to disease-specific care pathways in the context of multimorbidity.[15]

The aim of this study was to test and validate a new PROM, the Long-Term Conditions Questionnaire (LTCQ). The LTCQ is intended to be relevant and acceptable for people with single or multiple LTCs (physical and/or mental health conditions), and meaningful for health and social care providers in their capacities for monitoring and improving outcomes in LTCs. Additionally, the LTCQ is intended to be short, easy to interpret and feasible for use in different clinical settings. It is intended for use both as a tool for monitoring and enhancing individual care, and as a population-level tool for measuring health and social care performance, quality and outcomes. The scope of the LTCQ goes well beyond symptoms and physical function; its content development has been described previously and involved interviews with professional stakeholders,[16] qualitative in-depth interviews with people living with LTCs[17] and pretesting (eg, cognitive interviews, translatability assessment) to refine questionnaire items.[18]

## METHODS
Data were collected through two postal surveys (a main survey and a follow-up survey) administered to two cohorts: a healthcare sample recruited through primary care practices (data collected February 2016–July 2016), and a social care sample recruited through local authorities (LAs) that provide funding for social care services (data collected July 2016–January 2017). Methods and findings presented below follow STROBE reporting guidelines for cross-sectional studies[19] and COSMIN criteria for reporting measurement properties of health status questionnaires.[20] The latter indicates a minimum sample size of 10 subjects per questionnaire item (ie, 200 participants for this survey validation study), which exceeds the minimum of 100 subjects required for factor analysis within Classical Test Theory. Owing to the complexity of the construct being measured and the diversity of the target population, the study authors aimed to achieve a minimum sample size of 1000 participants.

### Participant recruitment
For the healthcare cohort, participants were recruited by 15 general practitioner (GP) practices from three regions of England (South East, North West, Yorkshire and Humber). In an effort to recruit a maximally diverse sample, the research team selected practices that served both rural and urban areas, and areas of high and low deprivation. For a participant to be invited into the study, the GP practice confirmed diagnosis of one of 11 specified LTCs: cancer within the last five years, chronic back pain, chronic obstructive pulmonary disease (COPD), diabetes, depression, irritable bowel syndrome (IBS), ischaemic heart disease, multiple sclerosis, osteoarthritis, severe mental health (as defined under the UK Quality and Outcomes Framework,[21] including psychoses, bipolar disorder and schizophrenia) and stroke. The 11 selected conditions were chosen by a panel of PROMs researchers and lay advisors, with the aim of maximising diversity in terms of symptoms, disease trajectory, prevalence, mean age of onset, likelihood of comorbidities, burden of disease, type of health and social care needed, level of self-management and burden of care. Each practice recruited from patient groups representing at least five of the 11 conditions, with some practices asked to prioritise certain conditions that were otherwise under-represented. Recruitment was restricted to those diagnosed more than 12 months previously to ensure that participants had adjusted to their diagnosis and had experienced a range of services and strategies for the management of their LTC(s). Only adults (ie, aged 18 years and above) able to consent who were able to communicate in English were included, with no upper age limit. A total of 2983 eligible patients were invited to participate for the healthcare cohort (approximately 200 study packs mailed out by each participating GP practice).

For the social care cohort, participants were recruited by four LAs of different types (unitary, metropolitan, county and London borough) in geographically diverse regions (North West, East of England, South West and Greater London) representing a mix of urban and rural communities. Individuals were eligible for the study if they received fully or partially funded long-term social care

support, provided that the primary reason for support was a physical disability, sensory impairment or a mental health condition as listed in table LTS001b on the Short and Long Term (SALT) mandatory data returns for social care.[22] Potential participants were eligible if they received community-based services, were at least aged 18 years and were able to communicate in English. Individuals who received nursing or residential care, whose primary reason of support was a learning disability or cognitive impairment (as listed on table LTS001b of SALT), or whose records indicated that they lacked mental capacity to consent to research were excluded. The research team provided each LA with study packs, which were mailed directly by the LAs to 2294 eligible participants. This was to ensure that no personal data of individuals were disclosed to the research team without consent.

## The surveys

The study packs contained an invitation letter from the GP/LA, a participant information sheet and the main survey (survey 1). Survey 1 included the Long-Term Conditions Questionnaire (LTCQ) and other measures for testing the LTCQ's construct validity: EQ-5D (5-level version including the EQ-VAS),[23] the Self-Efficacy for Managing Chronic Disease 6-item scale,[24] an Activities of Daily Living scale (ADLs)[25] and the Bayliss burden of morbidity scale (adapted with permission from the developers to include all conditions for which participants in this study had been recruited).[26] These were selected because they measure different domains that were hypothesised to underpin the LTCQ's broad construct of 'living well with LTCs': physical functioning, symptom burden and emotional well-being (EQ-5D); confidence to self-manage (Self-efficacy scale), functioning and independence (ADLs), cumulative impact of LTCs (Bayliss scale). Survey 1 also included demographic questions, questions on service use, a question about help needed to complete the questionnaire and a box for free-text comments. Additionally for the social care cohort, survey 1 included a measure of social care-related quality of life, the Adult Social Care Outcomes Toolkit,[27] but this measure is not included in the initial validation analysis for the total sample. A prepaid, addressed return envelope was provided in all study packs. After approximately two weeks, participants were sent a thank you/reminder letter from the GP/LA in order to encourage further responses.

Survey 1 contained an address slip through which respondents could express willingness to receive the follow-up survey (survey 2). Among those who provided contact details (n=980, 81%), a subsample of 693 respondents (57%) were sent survey 2 approximately two weeks after returning survey 1. The subsample included 54% of the healthcare cohort (n=499) and 66% of the social care cohort (n=194). Survey 2 contained only the LTCQ, a reduced number of demographics questions and a health transition question asking about changes in health status during the period between completing survey

1 and survey 2. A prepaid, addressed return envelope was provided with all questionnaires. Participants who had not returned survey 2 within approximately two weeks were sent a reminder letter.

## Analysis

All data were entered into SPSS (V.24), a statistical software package. A coding framework was specified in advance and used by all research team members for consistency in data entry. Data cleaning was undertaken via analysis of frequencies for all items in survey 1, with any anomalies checked against the original questionnaires and corrected as necessary. Particular attention was given to the 20 items of the LTCQ, for which any missing or multiple responses prompted visual inspection and verification/correction of data for the entire survey 1 questionnaire. The same procedure for data entry, checking and cleaning was followed for survey 2 among the health cohort, and for both survey 1 and survey 2 among the social care cohort, to ensure data quality across the full dataset.

Exploratory factor analysis of the 20 LTCQ items was undertaken (see the 'Results' section), from which it was concluded that the LTCQ could be scored as a single composite measure. The appropriateness of scoring items as a single scale was further evaluated through examination of inter-item correlations (acceptable if 0.8 or less) and item-total correlations (acceptable if 0.3 or more).[28] LTCQ items were scored on a scale from 0 (most negative response) to 4 (most positive response). Items 9–15 were negatively phrased and were therefore reverse-scored. Taking a conservative approach and following best practice guidelines,[29] only responses for which all 20 LTCQ items had been answered were included in the initial validation analysis. A sum of the 20 item scores was calculated and recalibrated to give an overall LTCQ score ranging from 0 to 100, with higher scores indicating a better level of 'living well with LTCs'. Cronbach's α was calculated as a measure of internal consistency of the scale. Test–retest reliability was assessed via calculation of the intraclass correlation coefficient (ICC) type 2 (two-way random effects, absolute agreement) among respondents who reported no change in health status between survey 1 and survey 2. Analysis of variance (ANOVA) was employed to compare LTCQ scores among subgroups within the sample (ie, by gender, age, health vs social care cohort, mental vs physical health conditions, number of conditions reported). Owing to the clustered study design (ie, participants recruited through selected GP practices and LAs), intracluster correlation coefficients (ICCCs)[30] were calculated for each item to assess the extent to which variance in responses was associated with recruitment site.

Scores for all existing measures were calculated according to developers' instructions. For the EQ-5D-5L, value sets recently reported for a population in England were used to calculate a single index value for each participant's reported health state[31]; scores are only calculated if all five items have been completed, with a theoretical range of −0.28 (a state worse than death) to 1 (best

**Table 1** Participant characteristics (n=1211)

| Response option | n (%) |
|---|---|
| Recruitment | |
| Healthcare (via General Practitioner practice) | 917 (76%) |
| Social care (via Local Authority) | 294 (24%) |
| Age (years) | |
| 18–49 | 162 (13%) |
| 50–64 | 277 (23%) |
| 65–74 | 331 (27%) |
| 75–84 | 259 (21%) |
| 85+ | 128 (11%) |
| (missing) | 54 (5%) |
| Gender | |
| Female | 656 (54%) |
| Male | 528 (44%) |
| (missing) | 27 (2%) |
| Condition reported* | |
| Depression/anxiety | 508 (42%) |
| Chronic back pain | 450 (37%) |
| Diabetes | 313 (26%) |
| Osteoarthritis | 308 (25%) |
| Colon problems (eg, irritable bowel syndrome) | 290 (24%) |
| Heart disease | 284 (24%) |
| Chronic obstructive pulmonary disease | 188 (16%) |
| Stroke | 185 (15%) |
| Cancer within the last 5 years | 166 (14%) |
| Bipolar/psychosis/schizophrenia | 88 (7%) |
| Multiple sclerosis | 75 (6%) |
| (missing) | 27 (2%) |
| Employment | |
| Retired from work | 554 (46%) |
| Permanently sick or disabled | 218 (18%) |
| Employed/full-time education | 211 (17%) |
| Doing something else (eg, volunteering) | 85 (7%) |
| Unemployed | 31 (3%) |
| (missing) | 112 (9%) |
| Marital status | |
| Married/civil partnership | 648 (54%) |
| Widowed | 224 (19%) |
| Divorced/separated | 168 (14%) |
| Single/never married | 144 (12%) |
| Ethnicity | |
| White British | 1097 (91%) |
| Other White (eg, Irish, European) | 38 (3%) |
| Black/Black British (eg, African, Caribbean) | 18 (2%) |
| Asian/Asian British (eg, Indian, Pakistani) | 17 (1%) |
| Mixed | 8 (0.6%) |
| (missing) | 33 (3%) |

Continued

**Table 1** Continued

| Response option | n (%) |
|---|---|
| Help needed completing questionnaire | |
| No help | 896 (74%) |
| Had help, but answers are my own | 227 (19%) |
| Someone answered for me (proxy) | 74 (6%) |
| (missing) | 14 (1%) |

*Figures add up to more than 100% because most participants reported multiple conditions.

possible health state). The EQ-VAS score, a measure of overall health on that day, ranges from 0 (the worst health you can imagine) to 100 (the best health you can imagine). For the Self-efficacy measure, six items asked about confidence in doing certain health-related activities on a scale from 1 (not at all confident) to 10 (totally confident); the overall score is calculated as the mean of item scores, provided that participants had completed at least four of the six items. The ADL score is calculated as the sum of all items for which difficulty in managing daily activities was reported, ranging from 0 (no difficulty with any listed activities) to 13 (at least some difficulty with all listed activities). The Bayliss burden of morbidity measure lists 25 LTCs and asks respondents to indicate the impact of each condition on their lives; a score of 0 indicates that the respondent does not have that condition, while scores for individual items ranging from 1 (has the condition but it does not limit daily activities at all) to 5 (has the condition and it limits daily activities a lot) indicates the impact of any reported condition. The total morbidity score was calculated as the sum of impact scores for all conditions reported, including up to three LTCs that respondents could list as 'other long-term conditions not mentioned above'. A count function was applied to the morbidity measure to calculate the number and type (physical or mental health) of LTCs reported by each respondent. For assessment of construct validity, correlations (Spearman's rho) were calculated for the LTCQ score in relation to all other measures.

## RESULTS
### Sample characteristics
A total of 917 participants were recruited through primary care (31% response rate), and 294 participants were recruited through social care (13% response rate), giving a total sample of 1211 participants (23% overall response rate). Demographic information is shown in table 1. The age range was 18–102 years, with a mean age of 67 (SD 15.3 years). Fifty-four per cent (n=656) were female, just over half were married or in a civil partnership (n=648, 54%) and just under half were fully retired from work (n=554, 46%). The sample was mainly white British (n=1097, 91%), with limited representation from non-white groups. Although participants were recruited on the basis of having one LTC, the sample exhibited a

**Table 2** LTCQ item responses (n=1211)

| Item | Never | Rarely | Sometimes | Often | Always | Missing | Not applicable† | ICCC+ |
|---|---|---|---|---|---|---|---|---|
| 1. Able to cope well with health conditions | 3% | 10% | 29% | 26% | 31% | 1.50% | | 0.059 |
| 2. Able to fulfil responsibilities | 12% | 15% | 20% | 18% | 32% | 1.90% | | 0.037 |
| 3. Able to be as physically active as you wanted | 21% | 20% | 21% | 17% | 20% | 1.20% | | 0.023 |
| 4. Felt in control of daily life | 8% | 13% | 22% | 21% | 35% | 1.30% | | 0.064 |
| 5. Able to take part in activities you enjoy | 16% | 22% | 22% | 17% | 21% | 1.40% | | 0.035 |
| 6. Felt that your home is suitable for your needs | 4% | 5% | 15% | 18% | **56%** | 1.50% | | 0.040 |
| 7. Felt safe at home | 2% | 4% | 10% | 19% | **64%** | 1.20% | | 0.028 |
| 8. Felt safe outside the home | 9% | 10% | 23% | 18% | 38% | 2.00% | | 0.060 |
| 9. Felt bothered by symptoms* | 9% | 12% | 33% | 26% | 19% | 1.50% | | 0.039 |
| 10. Felt more dependent on others than you wanted* | 19% | 13% | 20% | 21% | 27% | 1.00% | | 0.027 |
| 11. Felt lonely due to health conditions* | 34% | 15% | 24% | 15% | 11% | 1.00% | | 0.036 |
| 12. Worried about being treated differently* | 39% | 17% | 25% | 10% | 7% | 1.50% | | 0.053 |
| 13. Found health/other services difficult to cope with* | **20%** | 13% | 19% | 6% | 3% | 2.20% | 36% | 0.037 |
| 14. Found treatments difficult to cope with* | **31%** | 21% | 19% | 8% | 4% | 1.40% | **17%** | 0.033 |
| 15. Felt that your health conditions made you unhappy* | 20% | 16% | 31% | 18% | 13% | 1.30% | | 0.068 |
| 16. Felt you knew enough about your health conditions | 5% | 11% | 24% | 26% | 32% | 2.10% | | 0.005 |
| 17. Had enough social contact with people | 5% | 13% | 21% | 21% | 38% | 2.10% | | 0.052 |
| 18. Had enough support to cope well with health conditions | 4% | 9% | 21% | 25% | **40%** | 1.40% | | 0.060 |
| 19. Felt confident in managing health conditions | 6% | 8% | 23% | 22% | 40% | 1.20% | | 0.052 |
| 20. Able to live your life as you want | 16% | 17% | 20% | 18% | 28% | 1.00% | | 0.043 |

+ICCC for item responses across recruiting GP practices. A coefficient of 1 would indicate that all responses within a given cluster (practice) were identical, that is, all variance in responses is explained by cluster; coefficients approaching zero indicate negligible variance in responses across clusters (practices).

Bold values indicate ceiling effects for individual items.

*Questions 9–15 are reverse-scored, that is, 'never' is the most positive response option.

†Questions 13 and 14 have an additional response option: "have not received any health-related services/treatments in the past 4 weeks". For analysis these responses were coded as 'never'.

GP, general practitioner; ICCC, intracluster correlation coefficient; LTCQ, Long-Term Conditions Questionnaire.

**Table 3A** Comparison of LTCQ scores among subsamples (main survey)

| LTCQ—main survey | n | Mean | SD | SE | 95% CI | α | ANOVA |
|---|---|---|---|---|---|---|---|
| Total sample | 1082 | 65.1 | 23.0 | 0.70 | 63.7 to 66.5 | 0.95 | |
| **Cohort** | | | | | | | |
| Healthcare sample | 838 | 70.0 | 21.7 | 0.75 | 68.6 to 71.5 | 0.95 | F (1, 1080)=201.8 |
| Social care sample | 244 | 48.2 | 19.1 | 1.22 | 45.8 to 50.8 | 0.92 | p<0.001 |
| **Gender** | | | | | | | |
| Male | 482 | 68.5 | 22.6 | 1.03 | 66.4 to 70.5 | 0.96 | F (1, 1057)=19.8 |
| Female | 577 | 62.2 | 23.0 | 0.96 | 60.3 to 64.1 | 0.95 | p<0.001 |
| **Age (years)*** | | | | | | | |
| 18–64 | 413 | 59.7 | 23.3 | 1.15 | 57.5 to 62.0 | 0.95 | F (2, 1032)=27.4 |
| 65–84 | 525 | 70.2 | 21.7 | 0.95 | 68.3 to 72.0 | 0.95 | p<0.001 |
| 85+ | 97 | 60.2 | 22.6 | 2.30 | 55.7 to 64.8 | 0.95 | |
| **Morbidity†** | | | | | | | |
| 1 LTC | 60 | 76.5 | 21.3 | 2.74 | 71.0 to 82.0 | 0.94 | |
| 2–4 LTCs | 320 | 73.9 | 21.3 | 1.19 | 71.5 to 76.2 | 0.95 | F (3, 1057)=75.3 |
| 5–7 LTCs | 351 | 67.9 | 21.5 | 1.15 | 65.7 to 70.2 | 0.95 | p<0.001 |
| 8+ LTCs | 330 | 51.1 | 20.1 | 1.11 | 49.0 to 53.3 | 0.93 | |
| **Mental health** | | | | | | | |
| No mental health condition reported | 624 | 74.2 | 20.2 | 0.81 | 72.6 to 75.8 | 0.94 | F (1, 1080)=291.2 |
| At least one mental health condition reported | 458 | 52.7 | 20.8 | 0.97 | 50.8 to 54.6 | 0.94 | p<0.001 |

*Post hoc analysis (Tukey's HSD) indicated that LTCQ scores were significantly higher for the 65–84 years age group compared with both other age groups (p<0.001).

†Post hoc analysis (Tukey's HSD) indicated that LTCQ scores were significantly lower for those with 5–7 LTCs compared with those with one LTC (p<0.05) and compared with those with 2–4 LTCs (p<0.01). LTCQ scores were significantly lower for those with 8+ LTCs compared with all other groups (p<0.001).

α, Cronbach's α (internal consistency) for 20 LTCQ items among subgroup; ANOVA, one-way between-group analysis of variance of LTCQ scores; LTC, long-term conditions; LTCQ, Long-Term Conditions Questionnaire; n, sample size; mean, mean LTCQ score for subsample.

high degree of multimorbidity; 1124 participants (93%) reported having two or more conditions, with a mean of 6.2 LTCs (SD 3.8 LTCs) reported across the sample. Five hundred twenty-two participants (43%) reported at least one mental health condition, with the majority of these also reporting at least one physical LTC.

### Acceptability

The LTCQ was completed in full by 1082 participants, which enabled calculation of an LTCQ score for 89% of the sample. Table 2 summarises the content and survey 1 response rates for individual items. Levels of missing data were low and broadly uniform across items, ranging from 1.0% (item 10, dependence; item 11, loneliness; item 20, living life as you want) to 2.2% (item 13, services difficult to cope with). The low levels of missing data for all individual items indicate acceptability of the LTCQ within this diverse sample.

### Floor/ceiling effects

For the total sample responses were generally skewed towards positive answers, with ceiling effects (ie, <5% and >40% of respondents endorsing the most negative and positive response options, respectively)[28] observed in

five items of the LTCQ (items 6, 7, 13, 14 and 18—see table 2). Ceiling effects were most pronounced for item 6 (home suitability) and item 7 (safety at home). These items convey content that was identified during previous qualitative phases of research as especially important for social care users, who represent a smaller portion of the sample. While it is worth noting these item-level ceiling effects for their potential implications in population-level analyses, they are not in themselves problematic given the LTCQ's potential use for individual-level monitoring, where a key aim would be to identify and support the relatively smaller proportion of respondents who selected negative response options. No ceiling effect was observed for the measure as a whole (ie, 15% or more of respondents achieving the highest possible score),[20] as <4% of respondents scored 100 on the LTCQ.

### Factor analysis

The dataset's suitability for factor analysis was assessed via Bartlett's test of sphericity (highly significant, p<0.001), the Kaiser-Meyer-Olkin (KMO) measure of sampling adequacy (0.96) and measures of sampling adequacy (MSA) (>0.9 for each item). As indicated by the ICCCs

**Table 3B** Comparison of LTCQ scores among subsamples (follow-up survey)

| LTCQ—follow-up survey | n | Mean | SD | SE | 95% CI | α | ANOVA |
|---|---|---|---|---|---|---|---|
| Total sample | 492 | 65.5 | 23.4 | 1.05 | 63.4 to 67.5 | 0.96 | |
| **Cohort** | | | | | | | |
| Healthcare sample | 379 | 70.3 | 22.3 | 1.15 | 68.0 to 72.6 | 0.96 | F(1, 490)=82.4 |
| Social care sample | 113 | 49.2 | 19.3 | 1.82 | 45.6 to 52.8 | 0.92 | p<0.001 |
| **Gender** | | | | | | | |
| Male | 229 | 68.7 | 22.8 | 1.50 | 65.8 to 71.7 | 0.96 | F(1, 480)=8.0 |
| Female | 253 | 62.8 | 23.4 | 1.47 | 59.9 to 65.7 | 0.96 | p<0.01 |
| **Age (years)*** | | | | | | | |
| 18–64 | 184 | 58.8 | 23.8 | 1.76 | 55.3 to 62.3 | 0.96 | F(2, 469)=14.4 |
| 65–84 | 250 | 70.6 | 22.1 | 1.39 | 67.9 to 73.4 | 0.96 | p<0.001 |
| 85+ | 38 | 65.6 | 20.7 | 3.35 | 58.8 to 72.4 | 0.94 | |
| **Morbidity†** | | | | | | | |
| 1 LTC | 22 | 78.2 | 18.3 | 3.89 | 70.1 to 86.3 | 0.93 | |
| 2–4 LTCs | 157 | 76.2 | 19.9 | 1.59 | 73.1 to 79.4 | 0.95 | F(3, 482)=43.9 |
| 5–7 LTCs | 143 | 67.7 | 23.6 | 1.97 | 63.8 to 71.6 | 0.97 | p<0.001 |
| 8+ LTCs | 164 | 50.9 | 19.4 | 1.51 | 47.9 to 53.9 | 0.93 | |
| **Mental health** | | | | | | | |
| No mental health condition reported | 290 | 74.6 | 20.1 | 1.18 | 72.4 to 77.1 | 0.95 | F(1, 490)=144.2 |
| At least one mental health condition reported | 202 | 52.1 | 21.3 | 1.50 | 49.1 to 55.0 | 0.95 | p<0.001 |

*Post hoc analysis (Tukey's HSD) indicated that LTCQ scores were significantly higher for the 65–84 years age group compared with the 18–64 years age group (p<0.001).

†Post hoc analysis (Tukey's HSD) indicated that LTCQ scores were significantly lower for those with 5–7 LTCs compared with those with 2–4 LTCs (p<0.01). LTCQ scores were significantly lower for those with 8+LTCs compared with all other groups (p<0.001).

α, Cronbach's α (internal consistency) for 20 LTCQ items among subgroup; ANOVA, one-way between-group analysis of variance of LTCQ scores; LTC, long-term conditions; LTCQ, Long-Term Conditions Questionnaire; n, sample size; mean, mean LTCQ score for subsample.

reported for each item in table 2, clustering effects by practice were very low (ie, ICCC values <0.10 for all items); thus the results of factor analysis reported below were interpreted as reasonably free from potential bias that could theoretically occur due to clustering of responses by recruitment site.

Exploratory factor analysis was undertaken using principal axis factoring (PAF). Three factors were extracted via the Kaiser criterion (eigenvalue >1), which explained 66% of variance: factor 1 (eigenvalue 10.9, explaining 55% of variance), factor 2 (eigenvalue 1.2, explaining 6.0% of variance), factor 3 (eigenvalue 1.1, explaining 5.6% of variance). Overextraction of factors is a recognised problem using this method,[32 33] and examination of the scree plot suggested that only the first factor should be retained. This was confirmed by Parallel Analysis,[32] which showed only the first factor with an eigenvalue exceeding the corresponding value generated for a random data matrix of the same size (20 items x 1082 respondents). For the one-factor solution, 19 LTCQ items loaded onto the general factor at levels ranging from 0.58 (good) to 0.86 (excellent),[34] with item 16 (knowledge about health conditions) loading less strongly (0.35). This evidence supports the LTCQ being scored as a single scale.

To check against underextraction of factors, three-factor solutions (indicated by the Kaiser criterion) were also examined. With orthogonal (Varimax) rotation, all 20 items loaded onto one of the three factors with a minimum loading of 0.35 (item 16; all other items loaded at 0.49 or higher), and 14 items cross-loaded onto additional factor(s). When oblique rotation (Direct Oblimin) was applied, the same pattern of primary loadings was observed across the three factors, with two items (item 10, dependence; item 19, confidence to manage illness) also loading weakly onto another factor. In this analysis, the three factors correlated substantially with each other (r>0.6 for all factor combinations), suggesting difficulty with interpreting and labelling the factors as distinct subscales. An examination of items within factors indicated that each factor contained multiple concepts, for example, while factor 3 appeared to broadly group social and environmental influences on the management of LTCs, it contained conceptually distinct items on safety, suitability of the home and social support. This observation is consistent with the conceptual framework from which items were developed,[17] in which 15 distinct concepts underpinned the 20 items tested in the initial validation survey. The items within each factor were summed and

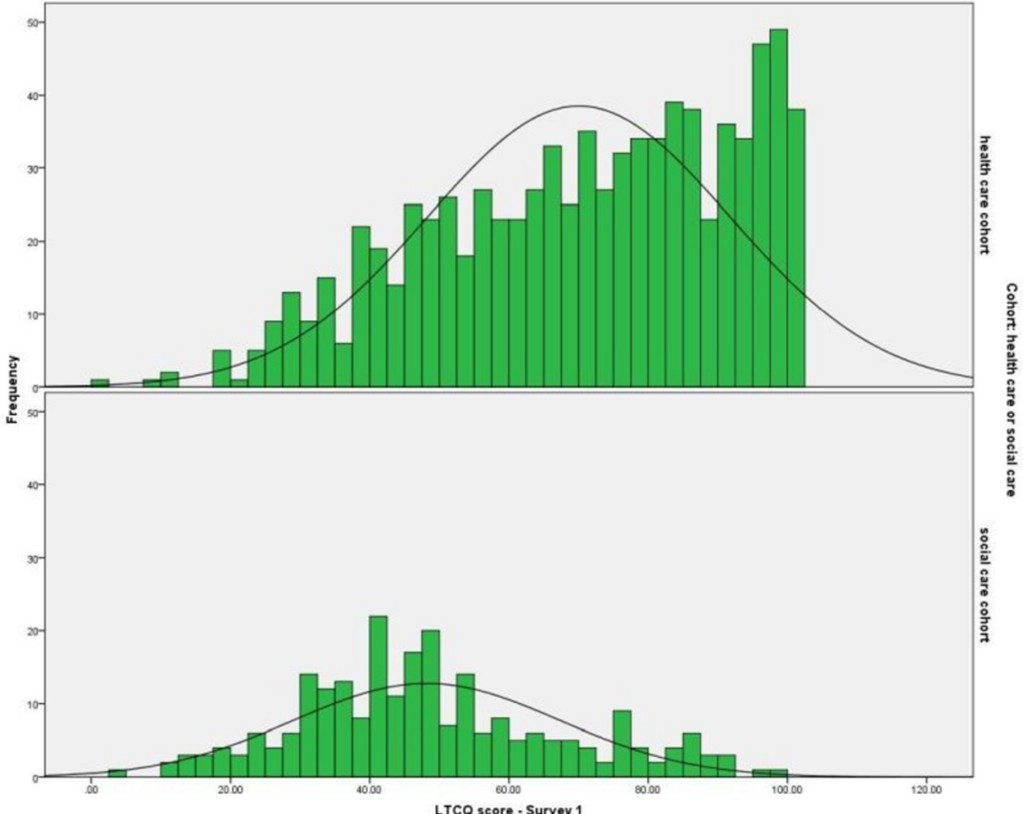

**Figure 1** Comparison of Long-Term Conditions Questionnaire (LTCQ) score distributions: healthcare versus social care cohorts.

calibrated to yield dimension scores ranging from 0 to 100, which were suitable for factor analysis (Bartlett's test highly significant, KMO and MSA values all >0.6). Higher-order factor analysis was undertaken using PAF and the Kaiser criterion; one factor was extracted with an eigenvalue of 2.3 that explained 75% of variance, with factor loadings of 0.93, 0.84 and 0.82. Thus, the appropriateness of scoring the LTCQ as a single composite measure that captures the broad construct of 'living well with LTCs' was confirmed.

### Internal consistency

The LTCQ exhibits high internal consistency across its 20 items (Cronbach's α=0.95). Corrected item-total correlations ranged from 0.35 (item 16, knowledge about health conditions) to 0.83 (item 4, felt in control of daily life), with negligible improvement in α if responses to item 16 were deleted. An examination of inter-item correlations showed that with one exception (item 3, able to be physically active and item 5, able to take part in enjoyable activities, r=0.83), associations between items were moderate rather than strong. No items were considered duplicates of other items and all items contributed substantially to the single scale; thus no items were deleted following initial analysis.

### Test–retest reliability

Of 693 participants sent the follow-up questionnaire, 544 (78%) completed and returned survey 2. LTCQ scores for the 383 participants (70%) who reported their health

as 'about the same' as two weeks ago were analysed for test–retest reliability. The ICC (type 2: two-way random effects, absolute agreement) for overall LTCQ scores between survey 1 and survey 2 was 0.94 (95% CI 0.93 to 0.95). Correlations for individual item responses between survey 1 and survey 2 were examined and found to be moderate or strong and significant for all items, ranging from 0.50 (item 16, knowledge about health conditions) to 0.83 (item 2, able to fulfil responsibilities). Frequencies of survey 2 responses were examined and found to follow the same pattern of skewing towards the most positive response options as for survey 1. Levels of missing data were similarly low (<2% missing for each item) for survey 2 as for survey 1, and high internal consistency of the scale (Cronbach's α=0.96) was again observed for survey 2 responses with complete LTCQ data (n=492).

### Subsample comparisons

The size and diversity of the sample enabled the comparison of LTCQ scores among different groups. Table 3A presents LTCQ score parameters and internal consistency measures for groups compared by cohort (health or social care), gender, age, number of conditions reported and presence or absence of a mental health condition. ANOVA confirmed statistically significant differences in mean scores in a predictable pattern: LTCQ scores were lower for the social care cohort, women, the youngest (aged under 50 years) and oldest (aged over 85 years) age

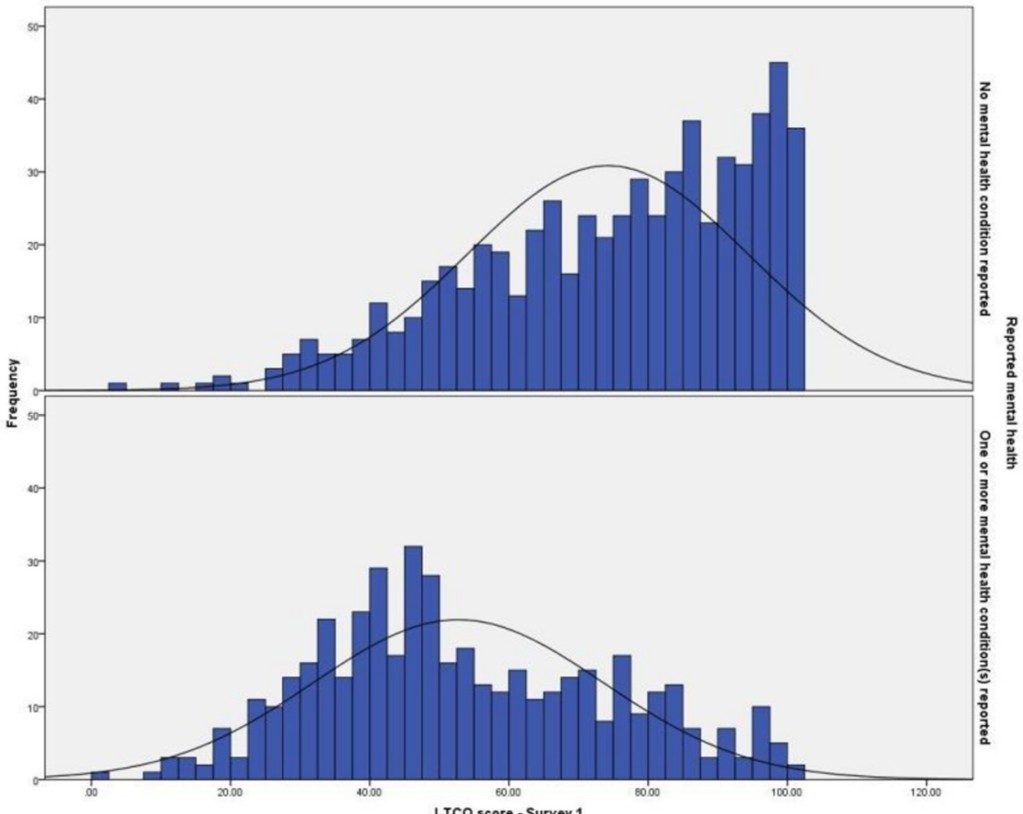

**Figure 2** Comparison of Long-Term Conditions Questionnaire (LTCQ) score distributions: absence or presence of reported mentalhealth condition(s).

groups, high multimorbidity (eight or more conditions reported) and the presence of at least one mental health condition. Internal consistency was high (Cronbach's α >0.9) for all groups. The analysis was repeated for responses to the follow-up survey (n=492), with the same pattern observed (see table 3B). Distributions of LTCQ scores by cohort (figure 1) and mental health (figure 2) are shown.

### Convergent construct validity

The sample's mean scores for the LTCQ, EQ-5D-5L, EQ-VAS, Self-efficacy scale, ADL scale and Bayliss burden of morbidity scale are shown in table 4, alongside correlations (Spearman's rho) of the LTCQ score with all other measures. Associations between the LTCQ and all reference measures were moderate to strong and in the expected directions, that is, positive for measures where higher scores indicated better outcomes (EQ-5D-5L, EQ-VAS, Self-efficacy scale), and negative for measures where higher scores indicated poorer outcomes (ADLs, Bayliss burden of morbidity). The collective strengths of association are notable given the different domains being captured across the measures (eg, physical and emotional functioning, confidence to self-manage, impact of LTCs on daily activities). Further item-level analyses are ongoing, but initial results suggest that while some LTCQ items correlate with specific items from reference measures, other LTCQ items seem to be tapping into distinct domains that underpin the broader construct of 'living well with LTCs'. For example, responses to LTCQ items reflecting personal autonomy (eg, items 1–5) are associated with responses to mobility, self-care and usual activities items from the EQ-5D; and LTCQ items reflecting illness burden (items 9–15) are associated with the EQ-5D depression/anxiety item; but LTCQ items reflecting social and environmental influences on the impact of LTCs (eg, social support, suitability of the home) are not strongly associated with items from existing measures. Taken together, this evidence indicates that the LTCQ score represents a more complex construct of 'living well with LTCs' that draws together domains from multiple existing measures.

### DISCUSSION

The LTCQ is a new generic PROM for capturing what it means to live well with long-term conditions. In this study, the LTCQ was found to be acceptable to a large and diverse sample of health and social care users (n=1211), with low levels of missing data across all items. For initial analysis, an LTCQ score was only computed if all items were completed; but given that 98% of the sample completed 18 items (90%) or more of the LTCQ, further work will explore the feasibility of imputing scores when one or two LTCQ items are missing. Internal consistency of the LTCQ is high, but analysis has not indicated direct repetition of

**Table 4** Construct validity

| Measure | Mean score (SD, SE, 95% CI) | Score range | Interpretation of higher score | Correlation with LTCQ score (Spearman's rho) |
|---|---|---|---|---|
| LTCQ | 65.1 (23.0, 0.70, 63.7 to 66.5) | 0–100 | Living better with long-term conditions | – |
| EQ-5D-5L | 0.62 (0.33, 0.01, 0.60 to 0.63) | −0.28–1 | Better health-related quality of life | 0.82*** |
| EQ-VAS | 62.4 (24.6, 0.72, 61.0 to 63.8) | 0–100 | Better health-related quality of life | 0.79*** |
| Self-efficacy scale | 6.2 (2.7, 0.08, 6.1 to 6.4) | 1–10 | Greater confidence for managing chronic disease | 0.87*** |
| Activities of Daily Living | 5.0 (4.8, 0.14, 4.7 to 5.3) | 0–13 | More problems with activities of daily living | −0.79*** |
| Bayliss burden of morbidity | 16.4 (13.1, 0.38, 15.7 to 17.2) | 0–150 | More limits on daily activities from LTCs | −0.64*** |

***p<0.001 (two-tailed)

content between items; this is consistent with the conceptual framework from which it was developed,[17] in which 15 distinct concepts underpinned the 20 items. Correlations with all reference measures (EQ-5D-5L, EQ-VAS, Self-efficacy scale, ADLs, Bayliss burden of morbidity) were strong and in the expected directions, supporting construct validity. Among this sample the LTCQ exhibited excellent test–retest reliability.

A strength of the study was the sample's diversity in the number, type and severity of health conditions reported, which indicates that the LTCQ is relevant for use across different types of LTCs. The potentially wide applicability of the LTCQ suggests that it could play a role in operationalising integrated person-centred care, with particular relevance for people with multimorbidity. That social care users have been specifically included in the sample is a further strength, suggesting that the LTCQ may be relevant for use in both health and social care settings. The range of reference measures used to validate the LTCQ is a third strength, demonstrating the complexity of 'living well with LTCs' that the LTCQ aims to measure, which is not fully captured by other existing measures.

Weaknesses of the study include the lower response rate achieved among the social care cohort and the relative homogeneity of the sample in terms of ethnicity. The response rate for the healthcare cohort (31%) was broadly in line with that of a previous pilot study[13] and other national health surveys.[8 35] The lower response rate for the social care cohort (13%), who reported lower levels of 'living well' in comparison to the healthcare cohort, may indicate less willingness or ability to engage with PROMs in comparison to other groups. These findings are not entirely unexpected given the similarity of this response rate to those of other projects assessing social care recipients.[36] Because the vast majority of participants in this study were white British, further testing is recommended to assess the relevance and acceptability of the LTCQ in other ethnic groups.

The LTCQ provides a more holistic approach to outcome measurement, encompassing but moving beyond the focus on symptoms and functioning seen in existing generic health status measures such as the EQ-5D. The strong correlations of LTCQ scores with both the EQ-5D and the Self-efficacy scale suggest that the broad construct measured by the LTCQ captures both functional abilities and self-confidence to manage illness, among other domains. The availability of a valid, generic measure for monitoring the cumulative impacts of LTCs could play a key role in facilitating the shift to new models of person-centred care. Crucial to emerging goals for redesigned services is individuals' capabilities for managing the many demands of living with LTCs. Equally important is the extent to which people have positive self-worth and are able to participate in society through meaningful and rewarding activities, including employment. In line with current policy,[37 38] a generic PROM for LTCs should also assess key aspects of relevance to social care including safety, control over life and quality of support—concepts that are included in the LTCQ. In drawing together a

unique range of health-related and social care-related items, the LTCQ fills a distinct gap in the availability of measures that are appropriate for evaluating integrated services in the context of multimorbidity.

A focus of future research will be to test the responsiveness of the LTCQ, which will be crucial for its potential use in routine monitoring. Furthermore, while this initial validation study has demonstrated the LTCQ's relevance for people with a diverse range of LTCs (including multimorbidity), further validation work is needed in populations not represented here, for example, those with dementia or learning difficulties, and those for whom English is not their first language. Translatability assessment of the LTCQ was undertaken during an earlier phase of its development[18] and concluded that it could be translated into multiple languages (eg, French, Polish, Arabic, Urdu, simplified Chinese). Following translation, the acceptability, validity and reliability of the LTCQ would need to be tested through further studies in non-English contexts. Further structural validation work, for example, employing Rasch analysis or bifactor models, would also contribute to the evidence base for this new measure.

## CONCLUSIONS

This paper provides encouraging evidence for the reliability and validity of the LTCQ, a new instrument for measuring 'living well' in the context of chronic illness. As a generic PROM that taps into a broad range of domains relevant for both health and social care settings, the LTCQ could meet a distinct need for holistic outcome measurement that facilitates integrated service provision. The measure's reliability among all subgroups within this diverse validation sample, coupled with previously reported evidence of content validity,[18] indicates that the LTCQ is relevant and acceptable for people with single or multiple LTCs, encompassing both physical and mental health conditions. In the context of increasing multimorbidity, a generic PROM that comprehensively captures what it means to live well with LTCs from the individual's perspective could support the implementation of person-centred care.

**Acknowledgements** The authors thank the participants with long-term conditions for taking part in this study, and all of the organisations who helped us to recruit participants. The authors thank Cheryl Hunter and Ann-Marie Towers for their contributions to the development of the LTCQ. The authors also thank Jane Dennett, Ed Ludlow and Alan Dargan for supporting the social care research team during recruitment.

**Contributors** RF, JEF, AC, CJ and MP conceived the study. RF and JEF secured its funding and managed its overall direction. JB and MB contributed to study design, ethics considerations and interpretation of study results as patient/public members of the research team. MP, LB and KJ led on securing ethics and other approvals for the study. AC, EG and CJ advised on data collection and data interpretation throughout the study. MP, CMP, LB, LG, CA, LK, DF and KJ were jointly responsible for participant recruitment (including working with participating organisations and developing the database search protocol) and for all aspects of data management (collection, entry, checking). CMP and MP led the analysis with direction from RF and CJ. CMP drafted the paper, which was critically reviewed by all authors. All authors contributed to revisions and approved the final version of the manuscript.

**Funding** This research was funded by the Policy Research Programme (PRP) in the Department of Health England, which supports the Quality and Outcomes of

Person-centred Care Policy Research Unit (QORU), and by the National Institute for Health Research (NIHR) Collaboration for Leadership in Applied Health Research and Care (CLAHRC) Oxford at Oxford Health NHS Foundation Trust. The views expressed are those of the authors and not necessarily those of the NHS, the NIHR or the Department of Health.

**Competing interests** Some authors had financial (salary) support from the two funding bodies: CMP, LK and EG from the NIHR via programme funding for CLAHRC Oxford, and LB, LG, DF, KJ and MP from the Department of Health England via programme funding for QORU policy research unit; the authors declare no financial relationships with any other organisations that might have an interest in the submitted work in the previous 3 years; the authors declare no other relationships or activities that could appear to have influenced the submitted work.

**Ethics approval** This study was reviewed by England's National Research Ethics Service (NRES) Committee East Midlands—Derby (reference 15/EM/0414). Approvals for the study were granted by the Health Research Authority of England's National Health Service (NHS), and local health and social care organisations linked to participant recruitment sites.

**Provenance and peer review** Not commissioned; externally peer reviewed.

**Data sharing statement** As stated in the approved study protocol, only members of the research team (ie, study authors) have access to the study data. The full anonymised data set was shared between all team members (University of Oxford and University of Kent). Direct access will be granted to authorised representatives from the sponsor or host institution for monitoring and/or audit of the study to ensure compliance with regulations.

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
