## [Reviewer comments · BMJ Open]

ARTICLE DETAILS

TITLE (PROVISIONAL)	The Long-Term Conditions Questionnaire (LTCQ): initial validation survey among primary care patients and social care recipients in England
AUTHORS	Potter, Caroline; Batchelder, Laurie; A'Court, Christine; Geneen, Louise; Kelly, Laura; Fox, Diane; Baker, Matthew; Bostock, Jennifer; Coulter, Angela; Fitzpatrick, Ray; Forder, Julien; Gibbons, Elizabeth; Jenkinson, Crispin; Jones, Karen; Peters, Michele

VERSION 1 - REVIEW

REVIEWER	Anita Slade Centre for patient reported outcomes research, University of Birmingham, UK
REVIEW RETURNED	13-Jun-2017

GENERAL COMMENTS	The paper is well written and you have attempted to present the data according to STROBE guidelines, however development of patient reported outcome tools should also follow COSMIN guidelines and the authors should review their recommendations for reviewing papers to enhance this publication and give it credence moving forward. There are a number of issues that you should address in order to make this paper suitable for publication. 1. Today people are moving away from CTT and using Rasch analysis as the most appropriate way of validating PRO. I think had this method been used it might have identified some issues relating to this tool and you would be advised to get help in validating this tool using that method or justify why you did not use this more robust method of validating your questionnaire. 2. While Cronbachs Alpha is often used, it is well known that there are issues with relying on this as a method of validation especially when you have a large sample size and large number of items its very likely you will have a large Cronbachs Alpha. The fact that you identified moderate rather than strong associations suggests your scale might not be unidimensional. Therefore I think you should use factor analysis to confirm that you do not have multi-dimensions in your scale (see COSMIN guidelines). This needs to be included in your publication as by your own admission you have a large enough sample size to do this therefore I do not understand why this was not presented. Given you have two distinct groups e.g. social care and health care recipients it might be useful to compare the results between the two groups given your sample size this should be feasible. I would suggest that potentially those receiving social care might be more disabled than the health care group and this would allow you to compare data between the two groups. ANOVA would also allow you to see if the measure could differentiate between the two
--

	groups. You could also present CA for the two groups as well. Missing data needs to include specific information on what you did, while you present the levels of missing data which is to be applauded it requires more information on how you dealt with it did you use a method of imputation if so what? Or did you simply leave incomplete data out of the analysis, this is not clear and needs more detail. Throughout you have only presented % you need to include actual numbers as well as percentages can be misleading on their own. For your test retest you need to state how many people i.e. what % were within the two week window as I note that reminders were sent two weeks later that's 4 weeks in total outside of the usual parameters for test retest. Within the group that reported changes within the time points did you look at responsiveness that might be a useful adjunct to the information collected. You need to report the standard error of measurement and confidence levels for the mean as well. While referencing the additional scales for convergent construct validity you do not give any information on these measures or why these were specifically chosen. You need to give a brief description on how the measures are scored. It would be useful to include direction of score in Figure 1 so you can easily compare the scores on each instrument. The fact that you state that your tool covers a number of domains also requires that you can justify taking a total score from the questionnaire and not the individual domains and this needs factor analysis to demonstrate that you have a unidimensional scale that warrants this approach. In table 3 I am not clear why you have correlated the number of LTCS with the scale, it might have been more useful to use this to compare scores on the LTCS to see if it could differentiate between different groups with different numbers of co-morbidities as you might expect. It would be useful to have more information on the characteristics of your population e.g. No of comorbidities, type of health problems, LTC if known. Comparisons of scores on the LTC by these groups would also add weight to your arguments that this is a valid tool for such a heterogeneous group. I hope you find these comments helpful.
--	---

REVIEWER	Ian Porter University of Exeter Medical School, UK
REVIEW RETURNED	20-Jun-2017

GENERAL COMMENTS	A well written concise paper. The aim of the study is clearly articulated - to validate the Long-Term Conditions Questionnaire (LTCQ) covering health and social care. The burden associated with long-term conditions is established, together with the need for Patient-reported outcome measures (PROMs) catering to this group to be short, easy to interpret, and which can be applied in different settings. The methods and analysis are succinctly outlined and evidence for the reliability and validity of the LTCQ is presented. As the authors highlight, there was low response rate among social care users, although this was not unexpected, and nevertheless the LTCQ is a promising measure with a wide range of potential applications in health and social care settings.
--

REVIEWER	Jan R. Boehnke University of Dundee, United Kingdom I am associate editor of a journal that covers exactly the topic area of this manuscript (Quality of life Research; sorry for the number of papers from that journal referenced here, but this kind of research falls exactly in our usual remit). I am in no way involved in any research in social care or primary care in the UK developing a competing instrument, neither commercial nor non-commercial.
REVIEW RETURNED	22-Jun-2017

GENERAL COMMENTS	The manuscript BMJOPEN-2017-017651 describes a study gathering preliminary empirical data on a new questionnaire tool to assess disease burden and impairment derived from long-term conditions in social care contexts. Based on this data, the authors' aim is to present some psychometric properties to establish whether future research for this tool might be promising and therefore in the long run, whether the use in practice can be recommended. A sample of N = 5277 potential respondents was approached in a multi-site multi-region sampling design, targeting both patients from 15 GP practices (i.e., primary care) as well as respondents receiving social care support at the moment. Of these, N = 1211 responded to the baseline mail survey and N = 544 responded to a follow-up mail survey to establish the LTCQ's re-test reliability. The information presented shows promising results for this collection of items. The paper is well written and the sampling design realised in this study is of particular high quality. The need for such an assessment instrument in social care is clearly established. Nevertheless, I think the level of detail regarding reporting of several elements of the LTCQ development is lacking detail and the psychometric results are too few in my opinion to be of use for either researchers or practitioners. And in this I appreciate that BMJOpen does not encourage a judgement on the significance of the study, but in my view the manuscript at the moment conveys too little detail to (a) provide reason for use of the instrument for research purposes; (b) enough detail for a use in practice settings that is on par with standards for the use of psychometric instruments; or (c) any data or theoretical information that can be used for robust testing in replication studies by other teams. I'll elaborate on these points in more detail below. 1) Report of Psychometric Properties I think my main point of criticism is that the authors state to present a psychometric validation of the LTCQ, but neither psychometric information about the scale development nor any psychometric analysis in line with external standards is presented. In particular: 1a) FACTOR ANALYSIS: Specifically the point about factor analyses is very pertinent. The authors specifically suggest that the LTCQ should be analysed as a single score. Although this is disguised as a "practicability" argument (p. 6), these are the only statistics that are presented, i.e. this paper will be cited as justification for using a single score although no justification for this is presented in the manuscript. And although the authors refer multiple times to factor analyses in their manuscript, no explanation is provided why they refrained from reporting the results of such. Several guidelines for reporting of psychometric data nevertheless suggest in various phrasings that this is a key criterion, either stating
---

explicitly that such analyses should be provided or implicitly asking for a justification for the suggested scoring system. Just to mention a few:

- The COSMIN checklist (most relevant for this exact research and practice area): Terwee et al., 2007, *J Clin Epi*, 60, 34-42
- The most elaborate detailed standard in QoL research which was elaborated in the context of item bank development, but for which most points are also applicable to HRQoL development in general: Reeve et al. 2007, *Medical Care*, 45, S22-31.
- Most relevant international document for all psychological testing, the Standards for educational and psychological testing published by the AERA, APA and NCME in 1999
- Reporting standards of the American Educational Research Association (AERA), *Educational Researcher*, 35, 33-40.
- And for a more general, embedded discussion: Carrig, M. M., & Hoyle, R. H. (2011). Measurement choices: Reliability, validity, and generalizability. In A. T. Panter & S. K. Sterba (Eds.), *Handbook of ethics in quantitative methodology* (pp. 127–157). New York: Routledge.

1b) Related minor point: The authors mention 'factor analysis' as one of their justifications for sample size in their methods section. But no factor analysis is performed in this paper, which seems incongruent.

1c) There are obviously many different approaches which the authors could use at this exploratory stage, but I will list a number of points that should be considered here. (i) If this was not already considered at inception of the instrument (and is not published already) a clear presentation of how many dimensions the instrument was originally developed to measure and whether it was conceived as a formative or a reflective assessment tool (e.g., Costa, 2015, *Quality of Life Research*, 24, 2057–2065) since both these aspects have consequences for the statistical strategy to test the items (factor analyses vs. principal component analysis; or for example network analyses as a relatively recent approach: Kossakowski et al., 2016, *Qual Life Res*, 25, 781). (ii) Depending on the first step most papers then present a mixture of exploratory and confirmatory analyses to provide tests of how well the intended structure actually describes item responses and to identify specific points of misfit (local stochastic dependencies, low loadings,... depending on approach) to derive hypotheses for further development and confirmatory testing. (iii) In these analyses an appropriate approach to deal with categorical item data should be used (e.g., Wirth & Edwards, 2007, *Psychological Methods*, 12, 58–79). (iv) If exploratory analyses are performed, robust approaches should be employed to determine the number of dimensions (e.g., parallel analysis, Hull method, or other). (vi) Since Cronbach's Alpha as well as most reliability coefficients do not provide information on the dimensionality of the instrument and are also not necessarily appropriate estimates of reliability (e.g., Sijtsma, 2009, On the use, the misuse, and the very limited usefulness of Cronbach's Alpha. *Psychometrika*, 74, 107–120; Gignac, 2014, *European Journal of Psychological Assessment*, 30, 130–139; Egberink & Meijer, 2011), *Assessment*, 18, 201–212) the report of that statistic alone does not justify the use of a single score. Additionally, if confirmatory factor analyses are deemed appropriate by the authors and used to explore dimensionality, such models allow for the calculation of more appropriate and informative coefficients.

Without elaborating far more on this point I think the analyses and

results presented so far in the manuscript are neither informative for researchers nor for practitioners, since the scores from instruments with unknown dimensionality (and relations of such scores with other data) cannot be interpreted (e.g., Smith et al., 2009, *Psychological Assessment*, 21, 272–284).

1d) **FAIRNESS:** Another aspect that is usually addressed in such first analyses is the potential for systematic bias in test scores against sub-populations of the intended target population. There is an ongoing debate about what a set of minimal criteria for such analyses could be, but especially for measures assessing health-related quality of life, variables such as gender and age (because of differential service use), and education (as proxy for certain health inequalities) can be seen as well-established (e.g., Teresi et al., 2009, *Psychology Science Quarterly*, 51(2) – online paper). For this particular application the mixture of samples from primary care and social care is another obvious candidate as well as the different long-term conditions (or at least multiple vs. no comorbidities). Without a more thorough analysis of this point it is unclear whether the LTCQ can be applied over a range of subpopulations without unfairly discriminating against one or more subpopulations. At least two broad methodological approaches exist for this, item response models (differential item functioning) or, more in line with the authors' current more classical test theory oriented approach, structural equation modelling (invariance testing).

1e) **CLUSTER SAMPLING:** It has been shown that even small cluster effects/ design effects have a detrimental effect on any statistical analysis as well as (if applied, see above) analyses aiming to correctly identify factorial structures (e.g. in more general cases, Pornprasertmanit et al., 2014, *Multivariate Behavioral Research*, 49, 518–543; Stochl et al., 2016, *International Journal of Methods in Psychiatric Research*, 25, 205–219). The authors do not take note of this in their analyses (especially for the 15 GP practices). I think as a minimum, the intra-class correlations per item should be reported (or their range in case they are low – e.g., < .05) and any analysis should take the cluster-sampling into account (which is good practice in survey methodology, e.g., Heeringa et al., 2010, *Applied Survey Data Analysis*, Chapman & Hall/CRC), including the correlations reported in table 3 which could be affected by this as well.

1f) **PRESENTING NORMS:** In the current version of the manuscript the authors implicitly present sample norms for the LTCQ by mentioning the sample average and its standard deviation in table 3. I am not sure whether something like this is intended by the authors, but presenting more detail on this would also be a way to develop the paper further. Although I am not sure whether this should be done at this stage since we practically know nothing about the psychometric appropriateness and functioning of this instrument (see arguments above), there is a certain tradition which would present more reference data since this is the kind of information that practitioners will need to interpret scores. For this, such a paper would present averages, standard deviations and reliability estimates by group, so that an individual's test result can be interpreted with reference to an appropriate comparison sample (. Most prominently this would probably be of interest for primary care vs. social care, gender and different LTCs depending on the sample sizes available within such categories. Also depending on sample size, sometimes norms for cross-tabulations of such categories

would be presented. This is obviously only needed in case relevant differences are found between samples. Numerous reports of such data are available, two recent examples are Huber et al., 2016, *Qual Life Res*, 25: 2787–2798 and Fat et al., 2017, *Qual Life Res*, 26: 1129–1144.

1g) Again, depending on what choices the authors make regarding analysing the structural validity of the LTCQ, bifactor models will allow a far more detailed assessment of construct validity than the analyses currently presented. Such analyses will allow to not only assessing which items go together and how much of the LTCQ score's variance is actually shared with other instruments. They will also provide insight into how much of the variance is actually specific to the LTCQ (the current correlations presented in table 3 treat all variance not shared with the LTCQ as 'statistical error', which is hopefully not correct), which is important because according to table 3, the LTCQ basically measures exactly the same thing as captured by EQ-5D utilities and the self-efficacy scale. Indicative references for such approaches: Gignac, 2014, *European Journal of Psychological Assessment*, 30, 130–139; Reise, 2012, *Multivariate Behavioral Research*, 47, 667–696; and Bonifay et al., 2017, *Clinical Psychological Science*, 5, 184–186.

MORE GENERAL POINTS

2) On page 3 of their manuscript the authors state "...where the objectives of health and social care services may be to maintain well-being..." The content discussed up to that point is about patient-reported outcome measures and all examples discussed measure health-related quality of life (which is the construct generic and specific measures in the PROMs category assess). These constructs are quite different from well-being (e.g., Stewart-Brown, 2013, *Defining and measuring mental health and wellbeing*. In L. Knifton & N. Quinn (Eds.), *Public mental health: global perspectives* (pp. 33–42). New York: McGraw Hill Open University Press). Further, the goal of person-centred care is described as "living well" (same page), which adds a third and again different aspect. I think the introduction makes a good case for the construct that is intended to be measured ('living well with LTC in a social care context'), but it is less clear how PROMs and HRQoL relate to it.

3) There is not much the authors can do about the unit-non-response in their survey, which is quite low, but for such a pilot study the benefits of having realised such an elaborate sampling design at all in my opinion clearly outweighs the lack of additional reminder surveys. Nevertheless, I am unsure why the missing data were not imputed at least for sensitivity analyses? A large amount of highly correlated information on quality of life and demographics was assessed in the survey and only unless respondents did not respond to any of those, the sample size could still be increased substantially in running such an analysis (non-response on individual items is clearly very low, table 2, but overall alone on the LTCQ the analysis loses 11% of the sample due to item non-response only: page 7). SPSS covers several approaches to do this and since especially with these highly correlated variables the missing-at-random assumption which most imputation approaches rely on is at least mildly plausible, this could be covered in more detail and a more robust approach could be used.

4) Page 9: "The LTCQ provides a more holistic approach to outcome measurement, encompassing but moving beyond the focus on

	symptoms and physical functioning seen in existing generic health status measures such as the EQ-5D." No analyses underpinning this claim are presented in this paper. The only related point reported here are the correlations with EQ-5D, Self-efficacy Scale and the ADL, which rather point to the fact that the LTCQ is measuring exactly the same thing. 5) Page 10: "...the LTCQ has potential for use in a variety of health and social care settings, as indicated by its initial validation in this diverse sample." I am not sure on which data/ results presented in the current paper this is based/ what it specifically refers to.
--	--

VERSION 1 – AUTHOR RESPONSE

Reviewer: 1

Reviewer Name: Anita Slade

Institution and Country: Centre for patient reported outcomes research, University of Birmingham, UK
 Competing Interests: None declared

The paper is well written and you have attempted to present the data according to STROBE guidelines, however development of patient reported outcome tools should also follow COSMIN guidelines and the authors should review their recommendations for reviewing papers to enhance this publication and give it credence moving forward.

COSMIN guidelines were followed in constructing and evaluating the measure. We have now made this more explicit (Methods, p.4) and have provided additional evidence, for example results of factor analysis (Results, p.7-8).

There are a number of issues that you should address in order to make this paper suitable for publication.

1. Today people are moving away from CTT and using Rasch analysis as the most appropriate way of validating PRO. I think had this method been used it might have identified some issues relating to this tool and you would be advised to get help in validating this tool using that method or justify why you did not use this more robust method of validating your questionnaire.

Both CTT and Rasch analysis are valid means of evaluating PROMs. Specific assumptions are applied for Rasch analysis, notably that items are rejected if they do not fit strict mathematical criteria for conforming to a unidimensional model. Given reviewers' comments about the need to test for uni-versus multi-dimensionality within the measure, we opted for CTT as an appropriate strategy for initial exploration of the LTCQ's underlying structure. On the basis of this initial analysis we have concluded that the LTCQ can be scored as a single scale (Results, p.8), so subsequent analyses (including Rasch analysis) of the measure are now in progress. However these subsequent analyses are not the focus of this paper.

2. While Cronbachs Alpha is often used, it is well known that there are issues with relying on this as a method of validation especially when you have a large sample size and large number of items its very likely you will have a large Cronbachs Alpha. The fact that you identified moderate rather than strong associations suggests your scale might not be unidimensional. Therefore I think you should use factor analysis to confirm that you do not have multi-dimensions in your scale (see COSMIN guidelines). This needs to be included in your publication as by your own admission you have a large enough sample size to do this therefore I do not understand why this was not presented.

Exploration of the scale using factor analysis had already been undertaken prior to the submission of this paper for review, and it had been our intention to publish these results subsequently (e.g. alongside results of the Rasch analysis currently in progress). However we take the reviewer's point that readers will want to be satisfied that this exploration of unidimensionality has already been undertaken prior to publication of first results from this initial validation study. A new section outlining

the approach and results from exploratory factor analysis has been added under Results (p.7-8).

Given you have two distinct groups e.g. social care and health care recipients it might be useful to compare the results between the two groups given your sample size this should be feasible. I would suggest that potentially those receiving social care might be more disabled than the health care group and this would allow you to compare data between the two groups. ANOVA would also allow you to see if the measure could differentiate between the two groups. You could also present CA for the two groups as well.

Differences in responses between these groups had already been explored but were not reported in the previous draft of the paper. A new table (Table 3, p.17-18) has been added that compares the LTCQ's performance among several groups within the sample: health versus social care cohorts, males versus females, age groups, the number of LTCs reported, and whether or not a mental health condition was reported. ANOVA indicated that the LTCQ could differentiate between sub-groups, as significant differences in mean LTCQ scores were found within each of the comparison groups. Cronbach's alpha is reported for each sub-group, for both the main and follow-up survey samples, in Table 3. High internal consistency was observed for all sub-groups including when sample sizes were small, for example among those reporting only one LTC who completed the follow-up survey (N=22).

Missing data needs to include specific information on what you did, while you present the levels of missing data which is to be applauded it requires more information on how you dealt with it did you use a method of imputation if so what? Or did you simply leave incomplete data out of the analysis, this is not clear and needs more detail.

As outlined in Methods – Analysis (p.6), no data imputation was undertaken for this initial validation analysis. Calculation of LTCQ scores and factor analysis were only undertaken for respondents with complete data for the measure. This is in line with best practice guidelines that highlight the underlying (untestable) assumptions and potential problems of any type of data imputation when constructing and testing new scales (e.g. Streiner and Norman 2008, Health Measurements Scales, pp. 139-140). We therefore took a conservative approach and did not impute data for missing items for this initial validation study. This conservative approach still yielded a large sample size (N=1082) of respondents with complete data, which was more than sufficient for comparing LTCQ scores among sub-groups and for undertaking exploratory factor analysis. Further analysis on the effects of data imputation for small numbers of missing LTCQ items is currently underway but is beyond the scope of this paper.

Throughout you have only presented % you need to include actual numbers as well as percentages can be misleading on their own.

Sample size numbers are included within the text (Results, p.7-9), within Table 1 (p.15) and Table 3 (p.17-18), and at the top of Table 2 (p.16). The authors opted to include only percentages within Table 2 for easier readability, but the numbers can be easily calculated from the sample size number included at the top of the table.

For your test retest you need to state how many people i.e. what % were within the two week window as I note that reminders were sent two weeks later that's 4 weeks in total outside of the usual parameters for test retest. Within the group that reported changes within the time points did you look at responsiveness that might be a useful adjunct to the information collected.

While a retest period of 2 to 14 days is considered usual, "expert opinions regarding the appropriate interval vary from an hour to a year depending on the task" (Streiner and Norman 2008:182). During pre-testing of the measure participants emphasized that variations in the impacts of LTCs could take much longer to manifest than for acute conditions, thus the recall period for answering questionnaire items is relatively longer (four weeks) than for some other measures, and it was assumed that for most participants health status would not change over that time scale. The main concern is that the health status of most participants does not change significantly over the test-retest interval; this was confirmed by the inclusion of a transition item in survey 2 that asked about health status now in comparison to two weeks ago (the interval between return of survey 1 and being sent survey 2), in which 70% of respondents said that their health in relation to their LTCs was 'about the same'. Only

those who reported no change in health status were included in the test-retest reliability analysis (Results, p.8).

Data from those who reported a change in health status across the test-retest interval imply that the measure detected those changes (i.e. those who reported improved health had an increase in mean LTCQ score and those who reported worse health had a decrease in mean LTCQ score). However as the follow-up study was designed to assess test-retest reliability rather than responsiveness, this evidence can only be viewed as suggestive. Responsiveness must be formally assessed in subsequent studies that are designed for that purpose.

You need to report the standard error of measurement and confidence levels for the mean as well.

These are now reported in Tables 3 and 4 (p. 17-19).

While referencing the additional scales for convergent construct validity you do not give any information on these measures or why these were specifically chosen. You need to give a brief description on how the measures are scored. it would be useful to include direction of score in Figure 1 so you can easily compare the scores on each instrument.

We have briefly elaborated in Methods - The surveys (p.5) on why these measures were chosen. Description of how each measure is scored is given under Methods – Analysis (p.6). Table 4 (p.19) summarizes the range and direction of scores for each measure to aid in comparability.

The fact that you state that your tool covers a number of domains also requires that you can justify taking a total score from the questionnaire and not the individual domains and this needs factor analysis to demonstrate that you have a unidimensional scale that warrants this approach.

As above, factor analysis is now presented under Results (p. 7-8) and justification is given as to why we concluded that the measure can be scored as a single scale.

In table 3 I am not clear why you have correlated the number of LTCS with the scale, it might have been more useful to use this to compare scores on the LTCS to see if it could differentiate between different groups with different numbers of co-morbidities as you might expect.

Number of LTCs has now been deleted from Table 4 (previously Table 3), and comparison of LTCQ scores for groups reporting different numbers of conditions is now given in Table 3 (new table, p.17-18).

It would be useful to have more information on the characteristics of your population e.g. No of comorbidities, type of health problems, LTC if known. Comparisons of scores on the LTC by these groups would also add weight to your arguments that this is a valid tool for such a heterogeneous group.

Table 1 (p.15) now includes the numbers of people reporting each of the 11 recruiting conditions. Table 3 (p. 17-18) gives mean LTCQ scores for sub-groups within the sample, including groups compared by number of comorbidities and mental versus physical health conditions.

I hope you find these comments helpful.

Thank you for these helpful comments, which have enabled us to present our study results with greater clarity.

Reviewer: 2

Reviewer Name: Ian Porter

Institution and Country: University of Exeter Medical School, UK

Competing Interests: None declared

A well written concise paper. The aim of the study is clearly articulated - to validate the Long-Term Conditions Questionnaire (LTCQ) covering health and social care. The burden associated with long-

term conditions is established, together with the need for Patient-reported outcome measures (PROMs) catering to this group to be short, easy to interpret, and which can be applied in different settings. The methods and analysis are succinctly outlined and evidence for the reliability and validity of the LTCQ is presented. As the authors highlight, there was low response rate among social care users, although this was not unexpected, and nevertheless the LTCQ is a promising measure with a wide range of potential applications in health and social care settings.

Thank you for these encouraging comments.

Reviewer: 3

Reviewer Name: Jan R. Boehnke

Institution and Country: University of Dundee, United Kingdom

Competing Interests: I am associate editor of a journal that covers exactly the topic area of this manuscript (Quality of life Research; sorry for the number of papers from that journal referenced here, but this kind of research falls exactly in our usual remit). I am in no way involved in any research in social care or primary care in the UK developing a competing instrument, neither commercial nor non-commercial.

The manuscript BMJOPEN-2017-017651 describes a study gathering preliminary empirical data on a new questionnaire tool to assess disease burden and impairment derived from long-term conditions in social care contexts. Based on this data, the authors' aim is to present some psychometric properties to establish whether future research for this tool might be promising and therefore in the long run, whether the use in practice can be recommended.

A sample of N = 5277 potential respondents was approached in a multi-site multi-region sampling design, targeting both patients from 15 GP practices (i.e., primary care) as well as respondents receiving social care support at the moment. Of these, N = 1211 responded to the baseline mail survey and N = 544 responded to a follow-up mail survey to establish the LTCQ's re-test reliability. The information presented shows promising results for this collection of items.

The paper is well written and the sampling design realised in this study is of particular high quality. The need for such an assessment instrument in social care is clearly established. Nevertheless, I think the level of detail regarding reporting of several elements of the LTCQ development is lacking detail and the psychometric results are too few in my opinion to be of use for either researchers or practitioners. And in this I appreciate that BMJOpen does not encourage a judgement on the significance of the study, but in my view the manuscript at the moment conveys too little detail to (a) provide reason for use of the instrument for research purposes; (b) enough detail for a use in practice settings that is on par with standards for the use of psychometric instruments; or (c) any data or theoretical information that can be used for robust testing in replication studies by other teams. I'll elaborate on these points in more detail below.

Thank you for your positive comments regarding the purpose of the instrument, achievement of the sampling design, quality of the written presentation and promising results. We have taken on board the need for further detail in reporting the psychometric properties of the measure, and we have incorporated your suggestions as far as possible within the space limits and in acknowledgement of the journal's broad clinical and public health audience. Some of the analyses that you suggest are beyond the scope of this paper and would be better suited for a more specialist psychometrics audience, but they are certainly relevant to us as we conduct subsequent analyses and develop the measure further.

1) Report of Psychometric Properties

I think my main point of criticism is that the authors state to present a psychometric validation of the LTCQ, but neither psychometric information about the scale development nor any psychometric analysis in line with external standards is presented. In particular:

1a) FACTOR ANALYSIS: Specifically the point about factor analyses is very pertinent. The authors specifically suggest that the LTCQ should be analysed as a single score. Although this is disguised as a "practicability" argument (p. 6), these are the only statistics that are presented, i.e. this paper will be cited as justification for using a single score although no justification for this is presented in the manuscript. And although the authors refer multiple times to factor analyses in their manuscript, no explanation is provided why they refrained from reporting the results of such. Several guidelines for

reporting of psychometric data nevertheless suggest in various phrasings that this is a key criterion, either stating explicitly that such analyses should be provided or implicitly asking for a justification for the suggested scoring system. Just to mention a few:

- The COSMIN checklist (most relevant for this exact research and practice area): Terwee et al., 2007, *J Clin Epi*, 60, 34-42
- The most elaborate detailed standard in QoL research which was elaborated in the context of item bank development, but for which most points are also applicable to HRQoL development in general: Reeve et al. 2007, *Medical Care*, 45, S22-31.
- Most relevant international document for all psychological testing, the Standards for educational and psychological testing published by the AERA, APA and NCME in 1999
- Reporting standards of the American Educational Research Association (AERA), *Educational Researcher*, 35, 33-40.
- And for a more general, embedded discussion: Carrig, M. M., & Hoyle, R. H. (2011). Measurement choices: Reliability, validity, and generalizability. In A. T. Panter & S. K. Sterba (Eds.), *Handbook of ethics in quantitative methodology* (pp. 127–157). New York: Routledge.

(As above to Reviewer 1) COSMIN guidelines were followed in constructing and evaluating the measure. We have now made this more explicit (Methods, p.4) and have provided additional evidence, for example results of factor analysis (Results, p.7-8). Exploration of the scale using factor analysis had already been undertaken prior to the submission of this paper for review, and it had been our intention to publish these results subsequently (e.g. alongside results of the Rasch analysis currently in progress). However we take the reviewer's point that readers will want to be satisfied that this exploration of unidimensionality has already been undertaken prior to publication of first results from this initial validation study. A new section outlining the approach and results from exploratory factor analysis has been added under Results (p.7-8).

1b) Related minor point: The authors mention 'factor analysis' as one of their justifications for sample size in their methods section. But no factor analysis is performed in this paper, which seems incongruent.

As above, factor analysis is now presented under Results (p.7-8).

1c) There are obviously many different approaches which the authors could use at this exploratory stage, but I will list a number of points that should be considered here. (i) If this was not already considered at inception of the instrument (and is not published already) a clear presentation of how many dimensions the instrument was originally developed to measure and whether it was conceived as a formative or a reflective assessment tool (e.g., Costa, 2015, *Quality of Life Research*, 24, 2057–2065) since both these aspects have consequences for the statistical strategy to test the items (factor analyses vs. principal component analysis; or for example network analyses as a relatively recent approach: Kossakowski et al., 2016, *Qual Life Res*, 25, 781). (ii) Depending on the first step most papers then present a mixture of exploratory and confirmatory analyses to provide tests of how well the intended structure actually describes item responses and to identify specific points of misfit (local stochastic dependencies, low loadings,... depending on approach) to derive hypotheses for further development and confirmatory testing. (iii) In these analyses an appropriate approach to deal with categorical item data should be used (e.g., Wirth & Edwards, 2007, *Psychological Methods*, 12, 58–79). (iv) If exploratory analyses are performed, robust approaches should be employed to determine the number of dimensions (e.g., parallel analysis, Hull method, or other). (v) Since Cronbach's Alpha as well as most reliability coefficients do not provide information on the dimensionality of the instrument and are also not necessarily appropriate estimates of reliability (e.g., Sijtsma, 2009, *On the use, the misuse, and the very limited usefulness of Cronbach's Alpha*. *Psychometrika*, 74, 107–120; Gignac, 2014, *European Journal of Psychological Assessment*, 30, 130–139; Egberink & Meijer, 2011), *Assessment*, 18, 201–212) the report of that statistic alone does not justify the use of a single score. Additionally, if confirmatory factor analyses are deemed appropriate by the authors and used to explore dimensionality, such models allow for the calculation of more appropriate and informative coefficients.

Without elaborating far more on this point I think the analyses and results presented so far in the manuscript are neither informative for researchers nor for practitioners, since the scores from instruments with unknown dimensionality (and relations of such scores with other data) cannot be interpreted (e.g., Smith et al., 2009, *Psychological Assessment*, 21, 272–284).

Two papers on earlier phases of the instrument's development have been published and are referenced in this paper. The paper on development of the conceptual framework underpinning items (Peters et al. 2016) is referenced several times in the text (e.g. Results p.8, Discussion p.10), to note that the LTCQ's 20 items were developed from 15 distinct concepts drawn from both health and social care. At the conceptual development phase those concepts were grouped under broad themes of 'impacts of LTCs', 'experiences of services and support', and 'self-care'. However it was not clear at that stage if these themes would emerge as scalable dimensions within the measure, so testing for multidimensionality was always part of the analysis plan. As outlined in the new factor analysis section under Results (p.7-8), parallel analysis was used to confirm the appropriateness of a single-factor solution for scoring the measure. With further analysis three groupings of items emerged that were highly correlated with each other. These groupings differed from the three previous themes, and each contained items reflecting several distinct concepts. Higher-order factor analysis of the three groupings, alongside examination of inter-item and item-total correlations, provided further evidence that the 20 LTCQ items can be scored as a single scale.

The aim of this paper was to present the first psychometric evidence of the LTCQ's validity and reliability to a broad clinical and public health audience who might be interested in testing the measure within their own practice and/or specific clinical populations. As such the results presented focus on the extent of missing data (acceptability), the stability of the measure (internal consistency and test-retest reliability), and whether or not the LTCQ seems to be capturing the range of domains that one might expect to comprise the broad construct of 'living well with LTCs' (construct validity). We have now provided results of exploratory factor analysis to assure readers that the basis of the scoring system is sound. Confirmatory factor analysis will be an important part of expanding the evidence base around this new measure, but it is beyond the scope of the current paper.

1d) FAIRNESS: Another aspect that is usually addressed in such first analyses is the potential for systematic bias in test scores against sub-populations of the intended target population. There is an ongoing debate about what a set of minimal criteria for such analyses could be, but especially for measures assessing health-related quality of life, variables such as gender and age (because of differential service use), and education (as proxy for certain health inequalities) can be seen as well-established (e.g., Teresi et al., 2009, *Psychology Science Quarterly*, 51(2) – online paper). For this particular application the mixture of samples from primary care and social care is another obvious candidate as well as the different long-term conditions (or at least multiple vs. no comorbidities). Without a more thorough analysis of this point it is unclear whether the LTCQ can be applied over a range of subpopulations without unfairly discriminating against one or more subpopulations. At least two broad methodological approaches exist for this, item response models (differential item functioning) or, more in line with the authors' current more classical test theory oriented approach, structural equation modelling (invariance testing).

A new table (Table 3, p.17-18) outlines performance of the LTCQ among sub-groups of the sample. Sub-groups have been compared along the lines of gender, age, recruitment through health versus social care, numbers of conditions, and presence or absence of a mental health condition. Educational data (or other proxies for socioeconomic status) were not collected from individual participants, but the sampling design included recruiting organisations that served different geographical locations representing the full range of index of multiple deprivation (IMD) scores. The mean LTCQ scores reported in Table 3 give a first reference point for how the LTCQ might be expected to perform among the different sub-groups, but as these are first results from a single validation study they should not be treated as population norms. Item-level analyses such as differential item functioning will be an important contribution as the evidence base for the LTCQ continues to grow, but it is beyond the scope of this paper.

1e) CLUSTER SAMPLING: It has been shown that even small cluster effects/ design effects have a detrimental effect on any statistical analysis as well as (if applied, see above) analyses aiming to correctly identify factorial structures (e.g. in more general cases, Pornprasertmanit et al., 2014, *Multivariate Behavioral Research*, 49, 518–543; Stochl et al., 2016, *International Journal of Methods in Psychiatric Research*, 25, 205–219). The authors do not take note of this in their analyses (especially for the 15 GP practices). I think as a minimum, the intra-class correlations per item should be reported (or their range in case they are low – e.g., < .05) and any analysis should take the cluster-sampling into account (which is good practice in survey methodology, e.g., Heeringa et al., 2010,

Applied Survey Data Analysis, Chapman & Hall/CRC), including the correlations reported in table 3 which could be affected by this as well.

While we appreciate this point being made, we are not convinced that this applies to our data or to the results presented in this paper. The references above describe bias introduced in confirmatory factor analysis (not undertaken for this paper) when multiple parties are involved in data collection (e.g. when a measure is interviewer-administered, the range of scores will reflect both variation in individual respondents and variation between different interviewers – all respondents interviewed by the same interviewer would constitute a cluster). In this study, the 15 GP practices served as participant identification centres for inviting people who had a diagnosis of at least one of the eleven specified LTCs. Once identified the GP practice mailed potential participants the study information and main survey, which was self-completed by respondents. All responses were completed independently using the same tool (i.e. the LTCQ), so it is not clear why the GP practices should be considered separate 'raters' in the sense of cluster analysis.

As outlined in Methods – Participant recruitment (p.4), the recruiting GP practices and Local Authorities were chosen as a means of achieving a maximally diverse study sample. Practices varied both in terms of area deprivation scores and in the patient populations from which they recruited. When we analysed LTCQ scores by practice, we were not surprised to observe a general trend of lower scores coming from practices based in areas of higher deprivation and/or practices that focused their recruitment on high-impact conditions (e.g. severe mental health, chronic back pain). However the factor analysis and comparison of LTCQ scores among sub-groups were conducted on the data for the total sample, which was designed to capture the diversity of the measure's target population (i.e. all people living with at least one LTC). If in the opinion of the reviewer the potential for cluster effects could seriously undermine the veracity of the results presented (response rates, internal consistency of the measure, etc.), then we would appreciate clearer guidance on how we should address this. As presented the issue seems to be more of a methodological debate that sits outside of this journal's focus.

1f) PRESENTING NORMS: In the current version of the manuscript the authors implicitly present sample norms for the LTCQ by mentioning the sample average and its standard deviation in table 3. I am not sure whether something like this is intended by the authors, but presenting more detail on this would also be a way to develop the paper further. Although I am not sure whether this should be done at this stage since we practically know nothing about the psychometric appropriateness and functioning of this instrument (see arguments above), there is a certain tradition which would present more reference data since this is the kind of information that practitioners will need to interpret scores. For this, such a paper would present averages, standard deviations and reliability estimates by group, so that an individual's test result can be interpreted with reference to an appropriate comparison sample (. Most prominently this would probably be of interest for primary care vs. social care, gender and different LTCs depending on the sample sizes available within such categories. Also depending on sample size, sometimes norms for cross-tabulations of such categories would be presented. This is obviously only needed in case relevant differences are found between samples. Numerous reports of such data are available, two recent examples are Huber et al., 2016, Qual Life Res, 25: 2787–2798 and Fat et al., 2017, Qual Life Res, 26: 1129–1144.

Reference scores and reliability estimates for the different groups are now presented in Table 3 (p.17-18). As above, these should be treated as first reference points from an initial validation study rather than as definitive population norms.

1g) Again, depending on what choices the authors make regarding analysing the structural validity of the LTCQ, bifactor models will allow a far more detailed assessment of construct validity than the analyses currently presented. Such analyses will allow to not only assessing which items go together and how much of the LTCQ score's variance is actually shared with other instruments. They will also provide insight into how much of the variance is actually specific to the LTCQ (the current correlations presented in table 3 treat all variance not shared with the LTCQ as 'statistical error', which is hopefully not correct), which is important because according to table 3, the LTCQ basically measures exactly the same thing as captured by EQ-5D utilities and the self-efficacy scale. Indicative references for such approaches: Gignac, 2014, European Journal of Psychological Assessment, 30, 130–139; Reise, 2012, Multivariate Behavioral Research, 47, 667–696; and Bonifay et al., 2017, Clinical Psychological Science, 5, 184–186.

Thank you for this suggestion, which we are exploring as we undertake further analyses regarding the LTCQ's structure. These analyses are beyond the scope of the current paper, but we acknowledge that they will be important as the evidence base around the LTCQ continues to build (Discussion, p.11).

It is incorrect to say that the correlations observed between scores of the LTCQ and other measures should be interpreted as the instruments measuring exactly the same thing. As part of our ongoing item-level analysis into the LTCQ's structure it appears that some LTCQ items correlate strongly with some EQ5D and Self-efficacy items, but that other LTCQ items capture concepts that are distinct from either of these existing measures (Results, p.9). In the Discussion (p.10) we argue that in correlating strongly with both instruments (which measure different constructs), the LTCQ seems to be encompassing a broader range of concepts that collectively constitute 'living well with LTCs'. Both points above are consistent with the LTCQ's conceptual development (Peters et al. 2016), for which the inclusion of items that both reflected and moved beyond the traditional domains of health-related quality of life was an explicit aim.

MORE GENERAL POINTS

2) On page 3 of their manuscript the authors state "...where the objectives of health and social care services may be to maintain well-being..." The content discussed up to that point is about patient-reported outcome measures and all examples discussed measure health-related quality of life (which is the construct generic and specific measures in the PROMs category assess). These constructs are quite different from well-being (e.g., Stewart-Brown, 2013, Defining and measuring mental health and wellbeing. In L. Knifton & N. Quinn (Eds.), Public mental health: global perspectives (pp. 33–42). New York: McGraw Hill Open University Press). Further, the goal of person-centred care is described as "living well" (same page), which adds a third and again different aspect. I think the introduction makes a good case for the construct that is intended to be measured ('living well with LTC in a social care context'), but it is less clear how PROMs and HRQoL relate to it.

As above, PROMs can and certainly do capture the traditional domains of HRQoL (e.g. mobility, symptom burden, physical and emotional functioning), which to some extent also include mental health and well-being (e.g. the depression/anxiety domain of the EQ5D). However this is not to say that PROMs must be limited to these traditional domains. More recently developed instruments measure domains that have not typically been considered aspects of HRQoL (e.g. self-management, empowerment, personal safety), but that nonetheless have clear relevance for a complex construct such as 'living well with LTCs'. The LTCQ aims to measure the latter and therefore encompasses both traditional HRQoL domains and less traditional ones, including domains that have particular relevance for the social care context. We have corrected instances in the text where the LTCQ's construct was incorrectly labelled 'well-being' in order to better distinguish between the two constructs.

3) There is not much the authors can do about the unit-non-response in their survey, which is quite low, but for such a pilot study the benefits of having realised such an elaborate sampling design at all in my opinion clearly outweighs the lack of additional reminder surveys. Nevertheless, I am unsure why the missing data were not imputed at least for sensitivity analyses? A large amount of highly correlated information on quality of life and demographics was assessed in the survey and only unless respondents did not respond to any of those, the sample size could still be increased substantially in running such an analysis (non-response on individual items is clearly very low, table 2, but overall alone on the LTCQ the analysis loses 11% of the sample due to item non-response only: page 7). SPSS covers several approaches to do this and since especially with these highly correlated variables the missing-at-random assumption which most imputation approaches rely on is at least mildly plausible, this could be covered in more detail and a more robust approach could be used.

(As above to Reviewer 1) As outlined in Methods – Analysis (p.6), no data imputation was undertaken for this initial validation analysis. Calculation of LTCQ scores and factor analysis were only undertaken for respondents with complete data for the measure. This is in line with best practice guidelines that highlight the underlying (untestable) assumptions and potential problems of any type of data imputation when constructing and testing new scales (e.g. Streiner and Norman 2008, Health Measurements Scales, pp. 139-140). We therefore took a conservative approach and did not impute

data for missing items for this initial validation study. This conservative approach still yielded a large sample size (N=1082) of respondents with complete data, which was more than sufficient for comparing LTCQ scores among sub-groups and for undertaking exploratory factor analysis. Further analysis on the effects of data imputation for small numbers of missing LTCQ items is currently underway but is beyond the scope of this paper.

4) Page 9: "The LTCQ provides a more holistic approach to outcome measurement, encompassing but moving beyond the focus on symptoms and physical functioning seen in existing generic health status measures such as the EQ-5D." No analyses underpinning this claim are presented in this paper. The only related point reported here are the correlations with EQ-5D, Self-efficacy Scale and the ADL, which rather point to the fact that the LTCQ is measuring exactly the same thing.

As above, the correlations between the LTCQ and other measures should not be interpreted as all instruments measuring exactly the same thing – rather, the instruments measure constructs that are associated and that therefore vary together in predictable ways. Thus, someone with greater confidence at managing their health (measured by the Self-efficacy scale) might be expected to report higher levels of physical and emotional functioning (measured by the EQ5D) which would reflect some aspects of living well with LTCs (measured by the LTCQ). The statement quoted above is supported both by the work previously published on the LTCQ’s conceptual framework and item development, and by the point that immediately follows in the same paragraph that stems from analysis of construct validity (Results p.9, Table 4 p.19): “The strong correlations of LTCQ scores with both the EQ-5D and the Self-efficacy scale suggest that the LTCQ captures both functional abilities and self-confidence to manage illness, among other domains.”

5) Page 10: "...the LTCQ has potential for use in a variety of health and social care settings, as indicated by its initial validation in this diverse sample." I am not sure on which data/ results presented in the current paper this is based/ what it specifically refers to.

This statement has been rephrased for greater clarity (Conclusions, p.11). This conclusion is drawn in light of the evidence that is accumulating from this study and from previously published work on the development of the LTCQ. Evidence of the value that stakeholders could place on a new health and social care measure (Hunter et al. 2015), the breadth and novelty of the construct that the LTCQ aims to measure (Peters et al. 2016), and content validity and translatability of the LTCQ (Kelly et al. 2016) have been previously presented. This initial validation study adds evidence of acceptability, internal consistency, test-retest reliability and construct validity amongst a diverse sample of health and social care users. Taken together this evidence is suggestive of the LTCQ’s potential for use in multiple settings relevant to the target population.

VERSION 2 – REVIEW

REVIEWER	Jan R. Boehnke University of Dundee, United Kingdom I am associate editor of a journal that covers exactly the topic area of this manuscript (Quality of life Research; sorry for the number of papers from that journal referenced here, but this kind of research falls exactly in our usual remit). I am in no way involved in any research in social care or primary care in the UK developing a competing instrument, neither commercial nor non-commercial.
REVIEW RETURNED	29-Aug-2017
GENERAL COMMENTS	The revision of the manuscript BMJOPEN-2017-017651 presents detailed responses in the letter and also some changes to the manuscript. Overall, I think the manuscript falls short of what could have been achieved and would be useful to readers especially with view to the psychometric validation of the scale, but I think at least now the choices are well-documented in the paper and readers can evaluate the manuscript and validity in far greater detail than before.

	I have one change that I would suggest that should be implemented, and that is addressing the multilevel structure of their data. As the publications referenced in my previous review show, even very small intra-class correlations can have a very detrimental effect on the identification of factor structures – this is therefore by far not only "a methodological debate", but could have practical consequences. And it is neither relevant whether clustering is introduced by centres, raters or other units nor is it relevant whether authors perform exploratory or confirmatory analyses. In the survey literature these effects are called "design effects", the "ecological fallacy" is a well-known effect, and it seems in those literatures quite standard to tackle them, so I am not sure why this would not be done in this case. 1) I think at least the intra-class correlations due to clustering in the centres for the items could be reported. This would be only one additional column in table 2 – or even just a note if the ICCs are all negligible, stating the maximum value observed. This way the potential effect would be acknowledged and readers would have information about the extent of the cluster effect. 2) The way to tackle this problem would be to perform multilevel factor analyses (e.g., Mplus) or to separate the observed correlations into intra-class correlations, the between-centre correlation matrix and the within-centre correlation matrix – and to perform the FA on the latter. The best documentation for the latter procedure is probably available through Bill Revelle's work on the R package psych, which can also be accessed through SPSS (which is the software the authors use). But while I think 1) should be done, I would leave 2) to the editor (as well as depending on the result whether cluster effects are present at all). Explanation of "No" responses on checklist: 7. If statistics are used are they appropriate and described fully? As delineated, I think more optimal analyses could have been performed (i.e. no), but they have been fully described (i.e. yes). 14. To the best of your knowledge is the paper free from concerns over publication ethics (e.g. plagiarism, redundant publication, undeclared conflicts of interest)? The authors state themselves in their letter that other publications will follow based on this data set addressing the same question. That it is difficult to reconcile with "best of your knowledge".
--	---

VERSION 2 – AUTHOR RESPONSE

In acknowledgement of the literature on design effects and Reviewer 1's concern about the potential for bias in the factor analysis results, we have now included Intra-Cluster Correlation Coefficients (ICCCs) calculated for each item, based on the recruiting practices, in Table 2. While statistically a category of intra-class correlation coefficients (ICCs), we anticipate that the journal's broad readership will be more familiar with ICC employed as a coefficient of reliability where values approaching 1 are sought; this is the type of ICC described in the COSMIN checklist, and that we employ in the paper as a measure of test-retest reliability. Therefore we have labelled the figures reported in Table 2 as

ICCC, where values approaching 0 are sought, in order to avoid confusion. As now reported in the revised paper (Results, factor analysis section), the ICC values were all low. Therefore the second analysis suggested by Reviewer 1 (multilevel factor analysis) does not strengthen the paper.

We hope that we have now sufficiently addressed Reviewer 1's remaining concerns. We look forward to hearing from you.

VERSION 3 – REVIEW

REVIEWER	Jan R. Boehnke University of Dundee, UK
REVIEW RETURNED	26-Sep-2017
GENERAL COMMENTS	No further comments.